behaviour

cognition, learning, schooling, shoaling, social information, social learning

**Author for correspondence:**
Mike M. Webster
e-mail: mmw1@st-andrews.ac.uk

# No evidence for individual recognition in threespine or ninespine sticklebacks (*Gasterosteus aculeatus* or *Pungitius pungitius*)

## Mike M. Webster and Kevin N. Laland

Centre for Biological Diversity, School of Biology, University of St Andrews, St Andrews, Fife KY16 9TF, UK

MMW, 0000-0001-9597-6871; KNL, 0000-0002-2457-0900

Recognition plays an important role in the formation and organization of animal groups. Many animals are capable of class-level recognition, discriminating, for example, on the basis of species, kinship or familiarity. Individual recognition requires that animals recognize distinct cues, and learn to associate these with the specific individual from which they are derived. In this study, we asked whether sticklebacks (*Gasterosteus aculeatus* and *Pungitius pungitius*) were capable of learning to recognize individual conspecifics. We have used these fish as model organisms for studying selective social learning, and demonstrating a capacity for individual recognition in these species would provide an exciting opportunity for studying how biases for copying specific individuals shape the dynamics of information transmission. To test for individual recognition, we trained subjects to associate green illumination with the provision of a food reward close to one of two conspecifics, and, for comparison, one of two physical landmarks. Both species were capable of recognizing the rewarded landmark, but neither showed a preference for associating with the rewarded conspecific. Our study provides no evidence for individual recognition in either species. We speculate that the fission–fusion structure of their social groups may not favour a capacity for individual recognition.

## 1. Introduction

The ability of animals to recognize others underpins the social organization of groups and shapes many of the interactions that take place within them. Occurring at a range of resolutions, recognition can take the form of broad categorizations, such as

discrimination between species and sexes, to differentiating kin from non-kin or familiars from unfamiliars, to the recognition of specific individuals [1–3]. Most researchers draw a distinction between class-level recognition and individual recognition [4,5]. Sherman *et al.* [4] identify two forms of class-level recognition. In the first, an animal learns characteristics or cues that are distinct to another individual and associates these with class-specific information about that individual. In the second, the animal compares the phenotype of another to internal templates associated with different classes. For 'true' individual recognition to occur an animal must be capable of perceiving cues that are distinct to an individual and matching these to a template representing that specific individual, leading it to express distinct behaviour towards that individual [4–6].

Research has revealed numerous examples of individual recognition, deployed in a variety of different contexts. Recognition via individually distinct facial markings in the paper wasp (*Polistes fuscatus*) allows for the stabilization of dominance hierarchies within colonies, minimizing aggressive interactions [7,8]. Male olive frogs (*Babina adenopleura*) recognize the individual vocal signatures of neighbouring territory holders and are less aggressive towards them than they are towards strangers [9]. This is an example of the 'Dear Enemy effect', characterized by reduced aggressive competition between neighbours once territorial boundaries are established. Emperor penguin chicks (*Aptenodytes patagonicus*) can recognize the call of their parents as they return to feed them, distinguishing them from among the background noise of calls within the colony [10]. Similarly, mothers and pups of the northern fur seal (*Callorhinus ursinus*) are able to recognize each other's calls among the other calls in their rookeries [11]. Bottlenose dolphins (*Tursiops truncatus*) produce individually specific signature whistles, the functions of which include maintenance of group cohesion and affiliative bonds [12–14]. Individual recognition can allow for the formation of complex social interactions and representations, including coalitions, reputation-tracking and monitoring of third-party interaction dynamics [15].

Both class-level and individual recognition have been documented in fishes [16]. Various species are capable of kin recognition [17–19] and they can discriminate between their own and others' offspring and eggs [20,21]. Habitat- and diet-derived cues also play a significant role in shaping recognition. Here, recognition seems to be based upon self-referent matching, with fish preferring to shoal with those from similar backgrounds or fed similar diets to themselves [22–35]. Habitat- and diet-derived cues are labile and shoaling preferences can alter over a timescale of hours as fish are moved between chemically different environments [25,28].

Just as in paper wasps [7], a number of fishes use individually variable facial cues to visual recognize individuals [36–38]. Evidence of individual recognition in fishes also comes from experiments that have looked at aggression between neighbouring territory holders, using experimental manipulations to demonstrate dear enemy effects via chemical [39], acoustic cues [40] and visual [41] cues. Guppies are able to learn to recognize familiar individuals with which they have been housed, with familiarity developing over a period of days to weeks [42,43]. Griffiths & Magurran [44] found that guppies which had lived in stable groups in isolated pools during the dry season preferred to shoal with others from the same pool, but only when the number of fish within the pool was low (up to 36 in their experiment). When the population of guppies in the pool was greater, this preference was absent, raising the possibility of an upper limit on the number of identities that can be learned. Griffiths & Magurran [44] argued that, if class-level recognition were operating here, the preference for pool-mates should have persisted regardless of pool population size, and the fact that it did not suggest instead that individual identities are learned. Ward *et al.* [26] found that under laboratory conditions, guppies were capable of recognition based upon both shared habitat and diet (class-level recognition), even when they had been housed in unconnected aquaria, and of recognition based upon direct experience (while controlling for habitat and diet experience). It is not clear whether recognition based upon direct experience constitutes individual recognition in this case, as the possibility remains that the receivers recognized individuals with which they have recently interacted at the class-level too. At the very least though, these findings imply that two forms of class-level recognition are important in the social behaviour of guppies. Using the same experimental design, Ward *et al.* [26] found no evidence for recognition based upon direct experience in threespine sticklebacks, implying that they were not able to recognize individuals in this way. They were able to discriminate between shoals based upon habitat- and diet-derived cues, and Ward *et al.* [26] suggest that the different social environments experienced by guppies and sticklebacks, imposed upon them by the ecological conditions that they experience, may have driven the differences seen in their behaviour.

In this study, we asked whether three- and ninespine sticklebacks were capable of learning to recognize other individuals. We have previously used these species as model organisms to explore the mechanisms, functions and evolution of social learning [45–48], including selective social learning [49–

51], in which animals are more likely to copy others under certain conditions, or where they are more likely to observe and copy the behaviour of specific individuals. Earlier work has demonstrated that the two species differ in their abilities to use some forms of social information [52,53], but see [48]. With a broader interest in the question of whether sticklebacks can learn to observe and copy specific individuals selectively, we first set out to determine whether the two species were capable of individual recognition. To do this, we used an experimental design in which fish were trained to associate the presentation of a green light with the delivery of a food reward close to one of two stimulus objects. We used two stimulus object treatments, one a pair of physical landmarks in the form of coloured plastic blocks and the other a pair of conspecifics held within permeable and transparent compartments. The food reward was always paired with only one of the two landmarks or conspecifics. The use of artificial landmarks allowed us to determine whether the fish were capable of learning general associations between an object's physical properties and a reward, something we expected them to be readily able to do, and served as a useful comparison for the potentially more demanding task of discriminating between two phenotypically similar conspecifics. If the fish learned an association between a reward and a landmark or conspecific then we expected to see them spend more time close to the rewarded stimulus compared to the unrewarding stimulus when the green light was switched on. We hypothesized that the fish would be capable of learning to recognize both the rewarded landmark and the rewarded conspecific, with the latter demonstrating a capacity for individual recognition. Given that we have previously seen differences in how the two species learn in some contexts [52,53] but not in others [48,53], we kept an open mind over whether we would see differences between what the two species would learn in this study.

# 2. Methods

## 2.1. Collection and housing

Several hundred of both species of stickleback were collected from the same location, Melton Brook, a small artificially straightened tributary of the River Soar in Leicester in March 2010. The fish were collected using dipnets and were transported to our laboratory at the University of St Andrews. In the laboratory, they were housed in single species groups of 15–20 fish in 45 l aquaria at 10°C on a 12 : 12 light cycle. Lighting was provided via LED strip lights mounted above each aquarium. Each aquarium contained a layer of sand and was equipped with an air powered filter. Plastic plants were provided for shelter and enrichment. The fish were fed daily with frozen bloodworms. They were held under these conditions for around one month until the experiment began. In the following experiment, we used adults of both species measuring 35–40 mm in body length. Fish that were in reproductive condition or that were obviously parasitized were not used. Following the completion of this experiment, the experimental subjects were retained in the laboratory for use in other projects.

## 2.2. Training

We used an experimental design in which test subjects were trained to associate green light with the provision of a food reward close to one of two stimuli. We performed two treatments. In the first treatment, the stimuli consisted of artificial plastic landmarks, one blue and one yellow. In the second treatment, we used two conspecifics held at either end of the aquarium as the stimuli. The use of abiotic stimuli alongside living conspecifics allowed us to determine whether the sticklebacks were capable of learning to associate the green light with a particular stimulus, even if they proved incapable of individual recognition. We used 30 fish of each species in each of the two treatments. A further 60 fish of each species were used as stimulus fish in the second treatment. Two of the focal threespine sticklebacks in the individual recognition treatment died during the course of the experiment, leaving 28 threespines in this treatment.

Fish were trained in three batches of 40 fish (each with 20 fish of each species), with 10 fish of each species in the landmarks treatment and 10 in the individual recognition treatment. Individual fish were moved to training tanks (45 × 30 × 30 cm, water depth 25 cm, figure 1). Each aquarium contained a 1 cm deep layer of fine sand. A perforated transparent plastic barrier (five 1 mm diameter holes cm$^{-2}$, Penn Plax brand) was placed 10 cm from either end of the tank, dividing it into three segments. The test subject was placed within the central segment. An air stone connected to an air pump was placed in the corner of each of the two outer segments. This aerated the water and helped to move the water through

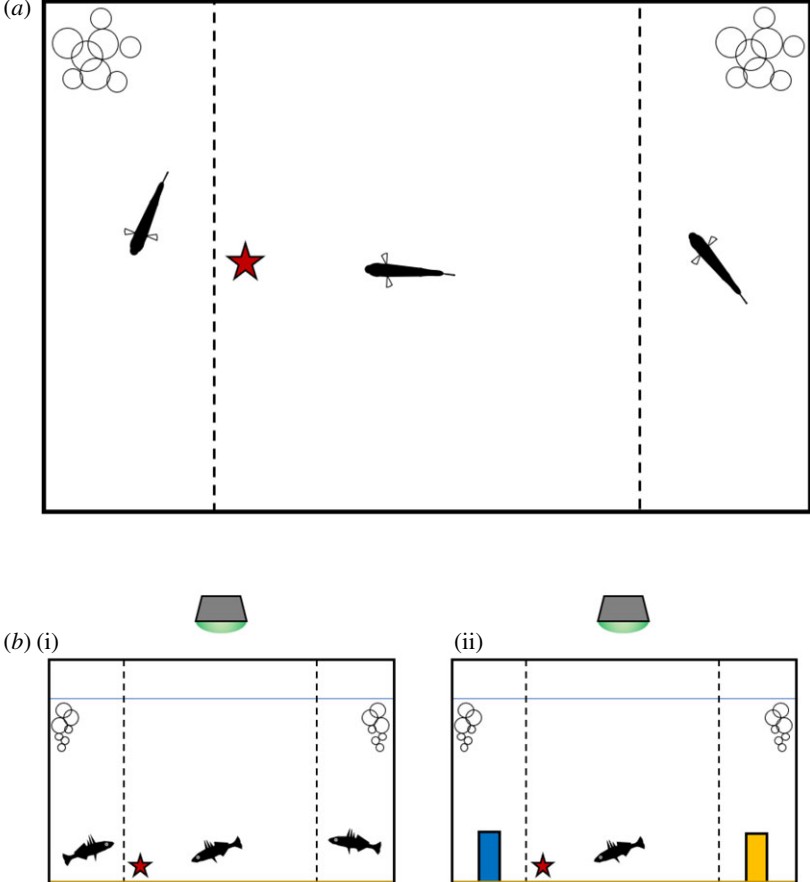

**Figure 1.** Diagram of the training tanks used in this experiment. (*a*) A plan view, depicting the test subject in the central chamber and stimulus fish housed in the smaller left and right chambers, behind perforated, colourless barriers (broken lines). The red star indicates the location of the food reward, such that, in this example, the left-hand fish is the rewarded stimulus (but note that the rewarded stimulus fish and landmarks were switched between the left and right side of the tanks in order to prevent the subject learning a side preference, as described in the main text). The bubbles indicate the locations of the air stones in each outer chamber, which served both to aerate the water and to facilitate water movement across the perforated barrier. (*b*) A side-on view of the same apparatus, depicting the treatments with stimulus conspecifics (i) and artificial landmarks (ii). The position of the green LED lamp is shown above the tank in this view.

the holes in the perforated barriers (confirmed with dye tests). In the landmarks treatments, a 48 mm tall 16 mm wide cuboid column was placed in the centre of each of the outer compartments. These were constructed from a stack of 5 standard 2 × 2 stud white Lego bricks. One was wrapped in yellow vinyl insolation tape, and the other in blue insulation tape (following [54]). Each landmark was anchored in position using a small amount of Plasticine, buried within the sand substrate, to prevent it from floating. In the individual recognition treatment, we used living conspecifics instead of artificial landmarks. A single fish was added to each end compartment. The stimulus fish were size matched to with two within 2 mm body length of one another and to the test subject. Within each tank, the test subject and two stimulus fish were each drawn from separate holding tanks and were initially unfamiliar to one another. Both species of stickleback are cryptically coloured and moreover are capable of changing colour via chromatophores found within their skin [55], so we made no attempt to quantify or match patterning. Each aquarium was screened with black paper placed on the outside of the tank, to minimize external disturbance. A lamp, 5 cm in diameter and containing nine LEDs (7000 k, 450 lx), was positioned 15 cm above each aquarium. A green filter was taped over each of the lamps.

The test subjects (and stimulus fish, in the individual recognition treatment) were held under these conditions for 3 days until the training began. During this period, each focal and stimulus fish was fed 10 bloodworms per day in a single feeding session. Fish were trained to associate green illumination with the provision of food close to one of the compartments holding a landmark or stimulus fish. We have used this training procedure in previous experiments and fish readily learn to

associate the light with food [56,57]. After the 3 day acclimation period, the training sessions began, and continued for a further 21 days. In the landmark treatment, one of the two landmarks, blue or yellow, was randomly designated the rewarded landmark. In the individual recognition treatment, one of the two stimulus fish was randomly selected in the same way. Two times per day, at 10.00 and 15.00, the green LED lamps above each tank were switched on. Sixty seconds after this, five blood worms were pipetted into the compartment housing the test subject, 1 cm from the barrier containing the rewarded landmark or conspecific. The experimenters' hands and arms were hidden from the fish by a screen, preventing them from learning to associate the experimenter with the arrival of the food. The green light was left switched on for a further 5 min. The test subjects typically consumed all of the food within this period. In the individual recognition treatments, the stimulus fish were fed 10 bloodworms each once per day at 16.30. Because we wanted to determine whether the test subjects were capable of learning to recognize a specific landmark or conspecific, we needed to rule out the possibility that they were simply learning the location the food reward. In order to do this, we randomized the location of the landmarks and conspecifics. Each day, each landmark or conspecifics was switched between or retained within its compartment with a probability of 0.5. This was performed separately for each replicate aquarium at 17.00 after the day's training and feeding were complete. Landmarks were pulled out and switched between compartments, and stimulus fish were carefully netted and switched in the same way. This took approximately 10 s and did not appear to cause any significant stress to the test subjects or stimulus fish. Where stimuli were retained within the same compartment between days, they were removed for several seconds and then placed back, in order to control for any disturbance. At this time, any uneaten food and faeces was carefully siphoned out of each compartment and the water levels topped up, as necessary.

## 2.3. Testing

Following this training period, the fish were tested. Fish were tested within their training tanks. A webcam was mounted above each tank allowing the location of the test subject to be recorded. The location of the rewarded and unrewarded stimuli was randomly determined 1 h before testing began. The test itself lasted 11 min. Within this period, the green light was switched on for 5 min and was left off for 5 min, with a 1 min period in between. The order of the light on/light off period was randomly determined for each trial. Each trial was filmed and we recorded the location of the test subject, whether it was with 5 cm of the barrier containing the rewarded or unrewarded stimulus, or within the 15 cm wide 'neutral' area at 10 s intervals, giving 30 point samples per period.

## 2.4. Statistical analysis

Within each testing period (light on and light off), we subtracted the number of point samples that the fish spent within 5 cm of the barrier containing the unrewarded stimulus from those spent within 5 cm of a barrier containing the rewarded stimulus, providing a time allocation score. These data were analysed using a general linear model with a Poisson distribution. Time allocation scores for the testing periods were treated as repeated measures. Treatment (landmark or conspecific stimulus), species (threespine or ninespine) and side of rewarded stimulus (left or right) were included as fixed factors, as was testing period order, that is whether the testing period with the light on came first or second. We included interactions between the testing period and treatment, testing period and species and treatment and species.

## 3. Results

The general linear model revealed significant effects of testing period ($F_{1,112} = 19.84$, $p < 0.001$, estimated power = 0.99, partial $\eta^2 = 0.15$), treatment ($F_{1,112} = 28.79$, $p < 0.001$, estimated power = 0.99, partial $\eta^2 = 0.21$) and a significant interaction between the two ($F_{1,112} = 11.51$, $p = 0.001$, estimated power = 0.92, partial $\eta^2 = 0.09$). This reflected the test subjects spending more time close to the rewarded artificial landmark than the unrewarded one in the testing period when the light was on, but not when it was off (figure 2). They showed no preference for the rewarded conspecific over the unrewarded one, either when the light was on or off. We saw no difference in the behaviour of the two species ($F_{1,112} = 0.17$, $p = 0.67$, estimated power = 0.07, partial $\eta^2 = 0.01$), nor any interaction between species and testing period or treatment ($F_{1,112} = 0.5$, $p = 0.69$; estimated power = 0.07, partial $\eta^2 = 0.01$; $F_{1,112} = 0.25$,

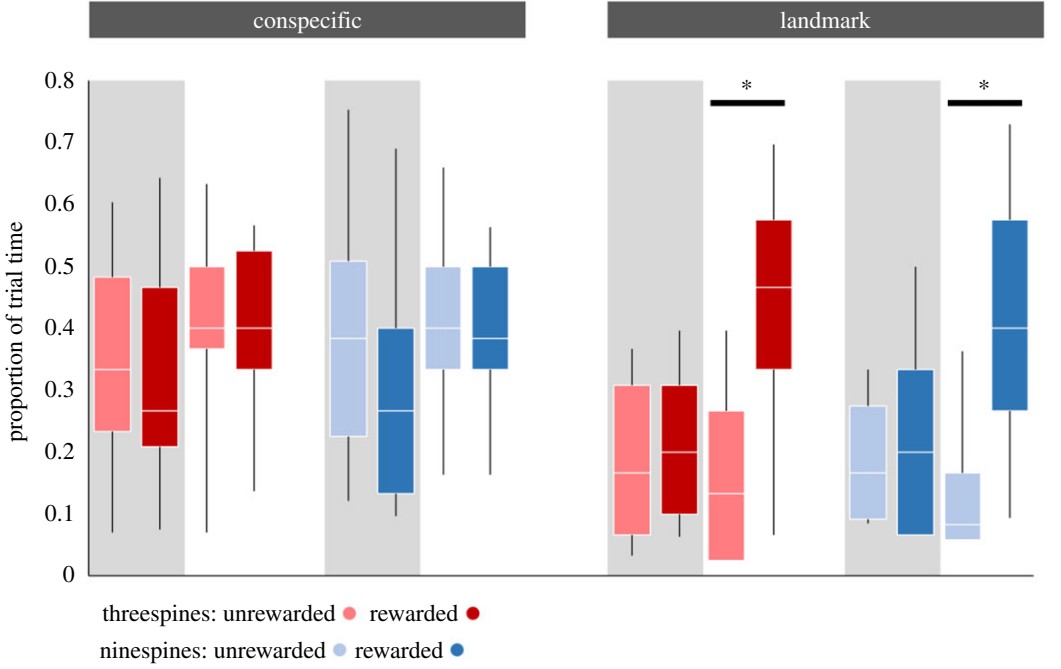

**Figure 2.** Pairs of boxplots displaying the proportion of trial time (median, interquartile range, 10th and 90th percentiles) spent by the test subject close to the rewarded and unrewarded conspecific/landmark when the conditioned stimulus, the green light was off (grey shaded) or on (no shading). Neither species of stickleback showed a preference for the rewarded conspecific either when the lights were off or on. Both showed a preference for the rewarded landmark but only when the lights were on, indicated by the asterisks. There were no differences between the two species.

$p = 0.61$, estimated power = 0.08, partial $\eta^2 = 0.02$). Testing period order had no effect upon test subject preference ($F_{1,112} = 0.03$, $p = 0.88$, estimated power = 0.05, partial $\eta^2 = 0.01$) and neither did the side of the rewarded stimulus ($F_{1,112} = 1.43$, $p = 0.23$, estimated power = 0.22, partial $\eta^2 = 0.02$). Within the landmark treatment group, there was no effect of landmark colour on preference, either before or after the green illumination was presented (before, colour: $F_{1,56} = 1.15$, $p = 0.22$; species $F_{1,56} = 0.07$, $p = 0.78$; interaction: $F_{1,56} = 1.27$, $p = 0.27$; after, colour: $F_{1,56} = 0.28$, $p = 0.60$; species $F_{1,56} = 0.02$, $p = 0.90$; interaction: $F_{1,56} = 1.03$, $p = 0.87$).

Note that the test subjects in the conspecifics treatment spent more time close to the conspecifics (rewarded or unrewarded, whether the green light was on or off) than the test subjects in the landmarks treatment spent close to the landmarks (in all cases except for the rewarded landmark when the light was on). This most likely reflects shoaling behaviour.

## 4. Discussion

This study found no evidence that either species of stickleback was capable of individual recognition. Both species were able to learn an association between the presentation of green illumination and the provision of a food reward close to an artificial landmark, but neither was able to learn to associate the green light with a particular conspecific. By including a treatment in which the fish were trained to associate a food award with an abiotic stimulus, we can rule out the possibilities that the fish were incapable of this form of associative learning generally, or that the design lacked the statistical power to detect a positive effect. Instead, our findings indicate that there was something about the stimulus of another conspecific that was hard to learn. The landmarks were consistent in appearance and differed consistently from one another, being different colours. By contrast, stimulus fish were selected to be phenotypically similar, in order to rule out the possibility of class-level recognition of, for example, larger versus smaller individuals. Many fishes, including sticklebacks [55], are capable of changing colour, and do so when stressed, to signal state and to match their background [58], which may have further increased the difficulty with which their appearance could be learned. It is possible that the landmarks were easier to discriminate than the conspecifics under the green light. This alone though does not seem a compelling explanation for the subjects' lack of any preference for the

rewarded conspecific, first because they had ample opportunity to inspect the stimulus fish before the green illumination was presented and second because chemical cues are known to be very important for class-level recognition in these species [24–26,28,29,32–35]. Chemical cues play an important role in recognition in many fishes [59] besides sticklebacks, but these can be labile and strongly affected by diet and the environment. In bullhead catfish (*Ictalurus nebulosus*), individuals recognize each other using chemical cues, with this recognition facilitating stable dominance relationships and reduced aggression between neighbouring territory holders. Experimental manipulation of the diet of one fish, however, can alter its chemical profile (measured through analysis of urinary free amino acids) such that it is no longer recognized by its neighbours, leading to heightened aggression [39]. Chemical cues used in class-level recognition by sticklebacks are similarly affected by changes in diet and habitat use (e.g. [25]). The absence of any apparent individual-specific characteristics, combined with the changeability of visual and chemical cues associated with individual fish in sticklebacks, may mitigate against the development of individual recognition. Indeed, individual recognition in many species seems to depend not merely upon the stability of the cues used by others for recognition but upon the evolution of individual-specific signals that specifically target receivers [5], a point that we return to below.

Our finding of no individual recognition in sticklebacks is consistent with that of [26], who showed that threespine sticklebacks showed shoaling preferences for conspecifics based upon class-level recognition, in this case self-referent matching of habitat- and diet-derived cues, but that they could not discriminate between conspecifics with which they had been housed and those from which they have been held separately, when habitat- and diet-derived cues were controlled for. By contrast, Milinski *et al.* [60] claim evidence for individual recognition in threespine sticklebacks. Studying cooperative predator inspection by groups of four fish, they found that particular pairs of sticklebacks tended to inspect the predator together more often than would be expected by chance. It is not clear from this study that the fish necessarily do recognize individual group mates however, and other explanations are possible. Positive social feedback between bolder fish that initiate movement out of cover and shyer fish that tend to follow reliably and which promote further forward movement in the bolder leaders as they follow them has been shown to drive collective movement in this species [61]. Potentially, a similar mechanism could drive paired inspections among the members of larger groups without the need for individual recognition to occur. Clearly, there is a need for further research into the importance of recognition in driving predator inspections by pairs of animals.

A capacity for individual recognition may depend both upon the ability of an observing individual to perceive and associate cues with the specific identity of a focal individual, but also upon the production of readily recognizable, individually specific signals by the focal individual [5]. These may include patterns or markings on the body [7,37,38], odours [62] or calls [13] that evolved to function as individual markers. Such signals might only evolve if there is a selective advantage to the signaller in being recognized, and to the receiver in recognizing them. Advantages may include reduction in the frequency or intensity of aggressive interactions between members of stable groups or between neighbouring territory holders [9,39], recognition between parents and offspring [10,11] or the facilitation of complex social interactions, such as coalition formation, reputation building, cooperation or tracking of third-party interaction dynamics [15], for example.

Many animals live in societies that are characterized by fission–fusion dynamics, where the groups regularly split or merge, or break down and reform. Individuals may interact with a large number of others from the local population, and under such conditions, there may be limited benefit in being able to recognize specific individuals, or constraints upon doing so imposed by the sheer number of others a given animal is likely to encounter [44]. Sticklebacks exhibit this kind of unstable, fission–fusion social system, with some short-term persistent interactions between individuals but also a high degree of mixing and turnover in shoal composition [63], which might explain why they have seemingly not evolved the capability to recognize conspecifics. Another possibility is that conspecifics are not recognized as a relevant cue that is associated with reward in a foraging context, but may be in other contexts. However, the fact that other studies have failed to find evidence for individual recognition in threespines in other contexts (e.g. shoaling [26]) leaves the evidence that these animals possess an individual-level recognition capability questionable at best.

Within fission–fusion groups, class-level recognition may be sufficient for effective social organization, allowing individuals to discriminate between different species and between con- and heterospecifics, between different phenotypes of their own species and between familiar and unfamiliar individuals [2]. Habitat- and diet-derived cues play an important role in shaping association preferences in both threespine and ninespine sticklebacks [23–26,28,29,32–35]), as well as in

other fishes [22,27,30,31]. Three- and ninespine sticklebacks spend more time associating with individuals that have been fed the same diet as themselves and which have been held in chemically similar water, implying self-referent matching of chemical profiles shapes these preferences. These cues are derived from amino acids [34], and potentially other sources. Chemical cues expressed via epidermal mucus, urine and faeces are known to convey information on relatedness, major histocompatibility complex profile and dominance in other fishes [39,64–67], and habitat- and diet-derived cues may also be transmitted this way. Field experiments have confirmed that such preferences can be affected by differences in water chemistry found between adjacent sites separated by just tens to hundreds of metres [25]. Moreover, recognition based upon self-referent matching of habitat and diet cues is extremely labile; when fish are moved from one environment to another, with different water chemistry, they acquire social preferences for fish from the new environment within a few hours [25,28]. Both threespine and ninespine sticklebacks form mixed species shoals in the wild [68], and display shoaling preferences for heterospecifics from the same habitat and diet background as themselves [24,30].

We were initially motivated to explore the question of individual recognition in these species because we were interested in the potential for individual recognition to shape selective social learning from specific individuals, 'who' social learning strategies [49]. Selective copying of high-ranking individuals has been documented in primates [69], and we were keen to investigate whether, for example, successful or innovative individuals might be disproportionately likely to be copied using the stickleback study system. Given their apparent inability to recognize individuals, this seems unlikely. However, researchers have had some success integrating class-level recognition into the study of selective social learning in sticklebacks, and in other fishes. Studying collective movement decisions, Sumpter *et al.* [70] found that when presented with pairs of artificial model conspecifics as potential leaders, threespine sticklebacks preferred to follow larger, stockier, paler and unblemished models over smaller, leaner, darker or speckled ones, suggesting discrimination and preference on the basis of better condition and health. When selecting between two socially demonstrated prey patches, ninespine sticklebacks were more likely to select the prey patch attended by larger conspecifics compared to one where smaller individuals had been fed [71]. Dugatkin & Godin [72] showed that younger female guppies were more likely to copy the mate choices of older females than younger ones, while older females were not swayed by the mate choices of younger ones. In both cases, the authors suggest that larger or older demonstrators might be more reliable than smaller or younger ones because they have been more successful or are more experienced. Finally, studying the diffusion of social information about prey patch locations, Atton *et al.* [32] created shoals of threespine sticklebacks in which half of the fish came from one laboratory population and half from another, such that each fish was familiar with about half of its shoal mates, and unfamiliar with the other half. While familiarly had only a weak effect upon social network structure within these groups, it was seen that fish were more likely to find prey patches by following familiar shoal mates than unfamiliar ones, indicating perhaps that they were paying more attention to their behaviour. Together, these examples demonstrate that class-level recognition can play a role in directing social learning, through a number of different behavioural mechanisms.

In summary, our study finds no evidence for individual recognition in either threespine or ninespine sticklebacks. We suggest that this probably reflects the unstable, fission–fusion social system of these species, which is characterized by low group fidelity and likely frequent encounters with large numbers of different individuals. Under such conditions, there may have been no pressure or no opportunity for individual recognition to develop, with class-level recognition sufficient to allow these animals to engage in a range of adaptive social behaviours.

Ethics. This study did not use humans or human tissues as subjects. This work was approved by the Animal Welfare and Ethics Committee at the University of St Andrews. No permit number was issued. Approval to collect fish from the wild and transport them to our laboratory was granted by the Animal Welfare and Ethics Committee at the University of St Andrews. No permit number was issued. All collections were carried out from publicly accessible lands.
Data accessibility. The data used in this study are provided in the electronic supplementary material.
Authors' contributions. M.M.W. designed the experiments and analysed the data, M.M.W. and K.N.L. wrote the manuscript.
Competing interests. The authors declare no competing interests.
Funding. This research was supported by ERC Advanced (EVOCULTURE 232823) and NERC (NE/D010365/1) grants to K.N.L.
Acknowledgements. We are grateful to Katherine Meacham for her assistance with this project.

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
