## [Reviewer comments · Royal Society Open Science]

Review History

RSOS-191703.R0 (Original submission)

Review form: Reviewer 1

Is the manuscript scientifically sound in its present form?

Yes

Are the interpretations and conclusions justified by the results?

No

Is the language acceptable?

Yes

Do you have any ethical concerns with this paper?

No

Have you any concerns about statistical analyses in this paper?

No

Recommendation?

Major revision is needed (please make suggestions in comments)

Comments to the Author(s)

The study investigates the ability of two species of fish for individual recognition. Replicates of 30 individuals of both species were trained to associate a green light with the provision of a food reward close to either an artificial landmark (blue or yellow cube) or one of two conspecifics in adjacent tank compartments. In the test, both species spend significantly more time near the rewarded landmark when the light was on, however, not next to the rewarded conspecific. The authors conclude that both species can learn the training procedure but are unable to individually recognise specific individuals.

The manuscript is well written, the methods are clear and the analysis appropriate, however I have one concern which needs clarification and possibly additional data:

The presence of conspecifics has been often shown to be an excellent reward in training individual fish and can also outweigh the importance of food rewards (especially when fish are readily fed twice a day 5 bloodworms). I believe it is difficult to tease apart if the fish with the conspecific stimuli didn't learn the association with a specific conspecific due to the lack of individual recognition or if they didn't show 'interest in the training' procedure as they had the constant reward of conspecifics nearby on both sides which might outweigh the food reward presented when the light was on. It is mentioned in the results that both species spend more time near the conspecific stimuli both when the light was on and off due to shoaling and not due to the prospect of the food reward. If fish had learned an association of the light with the food reward I would expect a change in behaviour, even though they haven't learned the correct conspecific stimuli? It would be good to look at specific behaviours when the light is on either during training for the first 60 seconds or over the whole 5 min during the test and compare them with behaviours present in the first 60 seconds training/5min test with the light on in the artificial stimuli treatment. Fish might show anticipatory behaviour of food if they did learn to associate the food with the light, e.g. faster swimming speed, swimming closer to the water surface, closer to the position where food is presented. In the present format there is no strong evidence of a lack of individual recognition but possibly a lack of interest in the training procedure.

Some minor questions:

Line 187. Why were the fish trained for 21 days? Is there evidence that this is the time they need for associative learning? Have fish been tested in between to see if they learned the association sooner?

Line 187-188. Why was this colour chosen?

In the discussion it was mentioned that sticklebacks have been shown to prefer individuals under better-condition and health as well as larger individuals over smaller ones (Lines 387-392). Is there data available about the condition of the stimuli fish (health, change in size or weight) over the time period of the study? A change in these conditions could maybe have impacted on the ability of the individuals to learn the association as more salient/preferred cues are contradicting it?

Review form: Reviewer 2

Is the manuscript scientifically sound in its present form?

Yes

Are the interpretations and conclusions justified by the results?

No

Is the language acceptable?

Yes

Do you have any ethical concerns with this paper?

No

Have you any concerns about statistical analyses in this paper?

No

Recommendation?

Major revision is needed (please make suggestions in comments)

Comments to the Author(s)

Sorry for being late, I found this study very interesting, therefore it took long time to read and understand. I have many questions and if I just didn't understand, apologize first.

1. There is no information that which sex was used as subjects and stimulus fishes. In no reproductive individuals, did not they have different interactions between males and females or among same sex?
2. Fishes were from same location but divided for one month in different aquariums. So, I ask that were the pairs of subjects and stimulus fishes from same aquariums or not, is there any influence on their behavior because of it and how was their interactions before the experiments if they were from same, did they have ranking in the aquarium?
3. Did subjects have enough motivation to get rewards as soon as possible? If stimulus fishes as themselves have stronger value rather than rewards, I think this experiment need to design again. Could subjects that did not get close to a rewarded stimulus fish at the moment green light was switching take rewards or get extra food after experiments?
4. Results showed subjects were always interested in both stimulus fishes, but there is no description of details in their social interactions. Getting closer includes some meanings and not only for shoaling. I think the behavior of stimulus fishes influence on the behavior of subjects. I ask to show their social interactions during experiments periods, is there any displays, aggressiveness, or modification?
5. Fishes, for example Medaka and Guppy, sometimes have a personal space just like a territory when it is bred in an aquarium alone or with few conspecifics. In this study, there is a possibility that subjects felt high threat-level against stimulus fishes that sometimes changed their location. A previous research in *Neolamprologus pulcher* (an African cichlid) revealed that territorial owners feel high threat against the neighbors that changed their location after dear enemy was established (Sogawa et al. 2016). If so, these results did not mean sticklebacks are incapable of individual recognition, these results probably mean they just had stronger motivation to watch out strange neighbors than the rewards.
6. In this study, the time subjects spent within 5 cm of the barrier was measured. There is no problem in landmark test by this measurement, because landmarks did not change their position. In contrast, stimulus fishes were, probably, always moving. In this case, the results should be measured as the time subjects spent within 5 cm of a stimulus fish. Because if subjects could associate rewards with stimulus fish, the distance of them would become important.
7. I think that even if subjects clear this test, it can't conclude stickleback is capable of individual recognition. There is a possibility that subjects just distinguish rewarded fish from unrewarded fish as a result of class level recognition. To show stickleback is capable of individual recognition, they must identify each rewarded fish. I think this study need to discuss this type of individual recognition called True Individual Recognition.
8. I also think that it can't conclude that stickleback is incapable of individual recognition or they could not associate rewards with stimulus fishes. I think previous studies do not support former conclusion, latter is easier to explain these results. Because fishes in the wild usually have

opportunity to associate food with location or landmark, but rare to associate food with individuals.

9. There is good discussion about chemical cues of fishes, but no discussion about individual recognition by sight. Some fishes are known that they can identify another by face pattern (e.g. Kohda 2015). Is there any previous study about individual recognition in stickleback by sight?

Review form: Reviewer 3

Is the manuscript scientifically sound in its present form?

Yes

Are the interpretations and conclusions justified by the results?

Yes

Is the language acceptable?

Yes

Do you have any ethical concerns with this paper?

No

Have you any concerns about statistical analyses in this paper?

No

Recommendation?

Accept with minor revision (please list in comments)

Comments to the Author(s)

This is a well-written, interesting manuscript about the lack of individual recognition in three-spined and nine-spined sticklebacks. It adds significantly to the literature of individual recognition in fishes. I enjoyed reading it and have some suggestions for improvement.

line 179: the green light is introduced here quite suddenly. The reader must be introduced about the reasons for using green light first. Add data on light intensity and light spectrum.

line 183: this is again unclear. In previous publications of the authors, fish were trained with a distinct green light stimulus but in the present manuscript it is the green illumination of the tank that served as stimulus, which maybe a more difficult learning task as test fish have to discriminate under green light conditions, especially between the two stimulus fish; the colour difference between the landmarks is huge in comparison even under green lighting. The association between award and illumination may be less evident than when the stimulus is a clear separate cue. This should be discussed. Mention how the tank was illuminated when the green light was not on.

line 204: netting is not the best way to transport fish and will cause stress even if not noticeable, or did you train fish to the netting?

Results: please test for side (left, right) bias; often there is a preference for one or the other side that must be controlled for. Please also test whether fish preferred one colour over the other.

Label the y-axis in Fig. 2.

line 291: "green light stimulus" is confusing as it was actually a green illumination

line 311: you did not measure in the ultraviolet waverange that sticklebacks can perceive well.

This should be discussed; there are fish that use UV signals for (species) recognition (Siebeck et al. 2010. *Curr Biol* 20:407-410). A paper on individual recognition based on visual cues in medaka was not cited (Wang & Takeuchi 2017. *eLife* 6:e24728).

line 347: write "split"

line 474, 476, 479: write "T.C.M." instead of "T.C."
References: write species names in italic

Decision letter (RSOS-191703.R0)

10-Feb-2020

Dear Dr Webster,

The editors assigned to your paper ("No evidence for individual recognition in threespine or ninespine sticklebacks (*Gasterosteus aculeatus* and *Pungitius pungitius*)") have now received comments from reviewers. We would like you to revise your paper in accordance with the referee and Associate Editor suggestions which can be found below (not including confidential reports to the Editor). Please note this decision does not guarantee eventual acceptance.

Please submit a copy of your revised paper before 04-Mar-2020. Please note that the revision deadline will expire at 00.00am on this date. If we do not hear from you within this time then it will be assumed that the paper has been withdrawn. In exceptional circumstances, extensions may be possible if agreed with the Editorial Office in advance. We do not allow multiple rounds of revision so we urge you to make every effort to fully address all of the comments at this stage. If deemed necessary by the Editors, your manuscript will be sent back to one or more of the original reviewers for assessment. If the original reviewers are not available, we may invite new reviewers.

- Data accessibility

It is a condition of publication that all supporting data are made available either as supplementary information or preferably in a suitable permanent repository. The data accessibility section should state where the article's supporting data can be accessed. This section should also include details, where possible of where to access other relevant research materials such as statistical tools, protocols, software etc can be accessed. If the data have been deposited in an external repository this section should list the database, accession number and link to the DOI for all data from the article that have been made publicly available. Data sets that have been

deposited in an external repository and have a DOI should also be appropriately cited in the manuscript and included in the reference list.

If you wish to submit your supporting data or code to Dryad (<http://datadryad.org/>), or modify your current submission to dryad, please use the following link:
<http://datadryad.org/submit?journalID=RSOS&manu=RSOS-191703>

- **Competing interests**

- **Authors' contributions**

- **Acknowledgements**

- **Funding statement**

Kind regards,

on behalf of Dr Kristina Sefc (Associate Editor) and Kevin Padian (Subject Editor)
openscience@royalsociety.org

Subject Editor's comments:

Thanks for your submission. We have received three thoughtful and constructive reviews, each of which brings out some important concerns about the design and presentation of the research. I will support a decision of "major revision" but you may find that it takes longer than our usual

timeline to address all of these concerns explicitly. If so, please request an extension from our editorial office. Best wishes.

Associate Editor's comments (Dr Kristina Sefc):

I would like to ask the authors to follow the suggestions of the three expert reviewers and look forward to a revision of the manuscript.

Subject Editor's comments:

Thanks for your submission. We have received three thoughtful and constructive reviews, each of which brings out some important concerns about the design and presentation of the research. I will support a decision of "major revision" but you may find that it takes longer than our usual timeline to address all of these concerns explicitly. If so, please request an extension from our editorial office. Best wishes.

Reviewers' Comments to Author:

Reviewer: 1

Comments to the Author(s)

The study investigates the ability of two species of fish for individual recognition. Replicates of 30 individuals of both species were trained to associate a green light with the provision of a food reward close to either an artificial landmark (blue or yellow cube) or one of two conspecifics in adjacent tank compartments. In the test, both species spend significantly more time near the rewarded landmark when the light was on, however, not next to the rewarded conspecific. The authors conclude that both species can learn the training procedure but are unable to individually recognise specific individuals.

The manuscript is well written, the methods are clear and the analysis appropriate, however I have one concern which needs clarification and possibly additional data:

The presence of conspecifics has been often shown to be an excellent reward in training individual fish and can also outweigh the importance of food rewards (especially when fish are readily fed twice a day 5 bloodworms). I believe it is difficult to tease apart if the fish with the conspecific stimuli didn't learn the association with a specific conspecific due to the lack of individual recognition or if they didn't show 'interest in the training' procedure as they had the constant reward of conspecifics nearby on both sides which might outweigh the food reward presented when the light was on. It is mentioned in the results that both species spend more time near the conspecific stimuli both when the light was on and off due to shoaling and not due to the prospect of the food reward. If fish had learned an association of the light with the food reward I would expect a change in behaviour, even though they haven't learned the correct conspecific stimuli? It would be good to look at specific behaviours when the light is on either during training for the first 60 seconds or over the whole 5 min during the test and compare them with behaviours present in the first 60 seconds training/5min test with the light on in the artificial stimuli treatment. Fish might show anticipatory behaviour of food if they did learn to associate the food with the light, e.g. faster swimming speed, swimming closer to the water surface, closer to the position where food is presented. In the present format there is no strong evidence of a lack of individual recognition but possibly a lack of interest in the training procedure.

Some minor questions:

Line 187. Why were the fish trained for 21 days? Is there evidence that this is the time they need for associative learning? Have fish been tested in between to see if they learned the association sooner?

Line 187-188. Why was this colour chosen?

In the discussion it was mentioned that sticklebacks have been shown to prefer individuals under better-condition and health as well as larger individuals over smaller ones (Lines 387-392). Is there data available about the condition of the stimuli fish (health, change in size or weight) over the time period of the study? A change in these conditions could maybe have impacted on the ability of the individuals to learn the association as more salient/preferred cues are contradicting it?

Reviewer: 2

Comments to the Author(s)

Sorry for beeing late, I found this study very interesting, therefore it took long time to read and understand. I have many questions and If I just didn't understand, apologize first.

1. There is no information that which sex was used as subjects and stimulus fishes. In no reproductive individuals, did not they have different interactions between males and females or among same sex?
2. Fishes were from same location but divided for one month in different aquariums. So, I ask that were the pairs of subjects and stimulus fishes from same aquariums or not, is there any influence on their behavior because of it and how was their interactions before the experiments if they were from same, did they have ranking in the aquarium?
3. Did subjects have enough motivation to get rewards as soon as possible? If stimulus fishes as themselves have stronger value rather than rewards, I think this experiment need to design again. Could subjects that did not get close to a rewarded stimulus fish at the moment green light was switching take rewards or get extra food after experiments?
4. Results showed subjects were always interested in both stimulus fishes, but there is no description of details in their social interactions. Getting closer includes some meanings and not only for shoaling. I think the behavior of stimulus fishes influence on the behavior of subjects. I ask to show their social interactions during experiments periods, is there any displays, aggressiveness, or modification?
5. Fishes, for example Medaka and Guppy, sometimes have a personal space just like a territory when it is bred in an aquarium alone or with few conspecifics. In this study, there is a possibility that subjects felt high threat-level against stimulus fishes that sometimes changed their location. A previous research in *Neolamprologus pulcher* (an African cichlid) revealed that territorial owners feel high threat against the neighbors that changed their location after dear enemy was established (Sogawa et al. 2016). If so, these results did not mean stickle backs are incapable of individual recognition, these results probably mean they just had stronger motivation to watch out strange neighbors than the rewards.
6. In this study, the time subjects spent within 5 cm of the barrier was measured. There is no problem in landmark test by this measurement, because landmarks did not change their position. In contrast, stimulus fishes were, probably, always moving. In this case, the results should be measured as the time subjects spent within 5 cm of a stimulus fish. Because if subjects could associate rewards with stimulus fish, the distance of them would become important.
7. I think that even if subjects clear this test, it can't conclude stickleback is capable of individual recognition. There is a possibility that subjects just distinguish rewarded fish from unrewarded fish as a result of class level recognition. To show stickleback is capable of individual recognition, they must identify each rewarded fish. I think this study need to discuss this type of individual recognition called True Individual Recognition.
8. I also think that it can't conclude that stickleback is incapable of individual recognition or they could not associate rewards with stimulus fishes. I think previous studies do not support former conclusion, latter is easier to explain these results. Because fishes in the wild usually have opportunity to associate food with location or landmark, but rare to associate food with individuals.

9. There is good discussion about chemical cues of fishes, but no discussion about individual recognition by sight. Some fishes are known that they can identify another by face pattern (e.g. Kohda 2015). Is there any previous study about individual recognition in stickleback by sight?

Reviewer: 3

Comments to the Author(s)

This is a well-written, interesting manuscript about the lack of individual recognition in three-spined and nine-spined sticklebacks. It adds significantly to the literature of individual recognition in fishes. I enjoyed reading it and have some suggestions for improvement.

line 179: the green light is introduced here quite suddenly. The reader must be introduced about the reasons for using green light first. Add data on light intensity and light spectrum.

line 183: this is again unclear. In previous publications of the authors, fish were trained with a distinct green light stimulus but in the present manuscript it is the green illumination of the tank that served as stimulus, which maybe a more difficult learning task as test fish have to discriminate under green light conditions, especially between the two stimulus fish; the colour difference between the landmarks is huge in comparison even under green lighting. The association between award and illumination may be less evident than when the stimulus is a clear separate cue. This should be discussed. Mention how the tank was illuminated when the green light was not on.

line 204: netting is not the best way to transport fish and will cause stress even if not noticeable, or did you train fish to the netting?

Results: please test for side (left, right) bias; often there is a preference for one or the other side that must be controlled for. Please also test whether fish preferred one colour over the other. Label the y-axis in Fig. 2.

line 291: "green light stimulus" is confusing as it was actually a green illumination

line 311: you did not measure in the ultraviolet waverange that sticklebacks can perceive well. This should be discussed; there are fish that use UV signals for (species) recognition (Siebeck et al. 2010. *Curr Biol* 20:407-410). A paper on individual recognition based on visual cues in medaka was not cited (Wang & Takeuchi 2017. *eLife* 6:e24728).

line 347: write "split"

line 474, 476, 479: write "T.C.M." instead of "T.C."

References: write species names in italic

Author's Response to Decision Letter for (RSOS-191703.R0)

See Appendix A.

RSOS-191703.R1 (Revision)

Review form: Reviewer 1

Is the manuscript scientifically sound in its present form?

Yes

Are the interpretations and conclusions justified by the results?

Yes

Is the language acceptable?

Yes

Do you have any ethical concerns with this paper?

No

Have you any concerns about statistical analyses in this paper?

No

Recommendation?

Accept as is

Comments to the Author(s)

The authors have clarified my concerns about individuals not learning about the conspecifics due to them already representing a better reward in comparison to the food. They have added some additional information in the discussion about this and have reworded the last couple of sentences in the abstract. I appreciate that the data is already more than 10 years old and more detailed analysis couldn't be conducted.

Overall I do really like the study, the set-up and design is very neat and the results are very clear. I believe the changes in the abstract and the addition in the discussion are appropriate and I would recommend the study for publication.

Review form: Reviewer 2

Is the manuscript scientifically sound in its present form?

Yes

Are the interpretations and conclusions justified by the results?

No

Is the language acceptable?

Yes

Do you have any ethical concerns with this paper?

No

Have you any concerns about statistical analyses in this paper?

No

Recommendation?

Accept with minor revision (please list in comments)

Comments to the Author(s)

I thank you for your revisions. I'm sorry for late to check your paper because of the influence of such situation around the world. You suggest your results showed that stickleback is not capable of individual recognition because of fission-fusion social system. But males make a territory in breeding season, is there dear enemy phenomenon? Can they modify their brain structure according to the season? And it can't conclude that stickleback is not capable of individual recognition through this experimental procedure. I think these results just showed it is difficult for them to learn to associate food with another individual or they are not capable of individual recognition during non-breeding season.

Review form: Reviewer 3

Is the manuscript scientifically sound in its present form?

Yes

Are the interpretations and conclusions justified by the results?

Yes

Is the language acceptable?

Yes

Do you have any ethical concerns with this paper?

No

Have you any concerns about statistical analyses in this paper?

No

Recommendation?

Accept as is

Comments to the Author(s)

I am happy with the revision. The authors have taken into account my comments and those of the other reviewers. There is one minor point that the authors may want to include in the discussion. Reviewer #2, comment 4 and 5, suggested the importance of territorial defense. The authors denied such behaviour in sticklebacks other than reproductive males. This is not correct. Juvenile (males and females) three-spined sticklebacks may defend territories (Bakker, T. C. M. & Feuth-de Bruijn, E. (1988). Juvenile territoriality in stickleback *Gasterosteus aculeatus* L. *Anim. Behav.* 36: 1556-1558.). Juveniles and adult females may also be aggressive towards conspecifics and one can artificially select for higher and lower aggression levels (Bakker, T. C. M. (1986). Aggressiveness in sticklebacks (*Gasterosteus aculeatus* L.): a behaviour-genetic study. *Behaviour* 98: 1-144.).

Decision letter (RSOS-191703.R1)

Dear Dr Webster:

On behalf of the Editors, I am pleased to inform you that your Manuscript RSOS-191703.R1 entitled "No evidence for individual recognition in threespine or ninespine sticklebacks (*Gasterosteus aculeatus* and *Pungitius pungitius*)" has been accepted for publication in Royal Society Open Science subject to minor revision in accordance with the referee suggestions. Please find the referees' comments at the end of this email.

The reviewers and Subject Editor have recommended publication, but also suggest some minor revisions to your manuscript. Therefore, I invite you to respond to the comments and revise your manuscript.

- Ethics statement

- Data accessibility

If you wish to submit your supporting data or code to Dryad (<http://datadryad.org/>), or modify your current submission to dryad, please use the following link:
<http://datadryad.org/submit?journalID=RSOS&manu=RSOS-191703.R1>

- Competing interests

- Authors' contributions

- Acknowledgements

- Funding statement

Because the schedule for publication is very tight, it is a condition of publication that you submit the revised version of your manuscript before 19-Jun-2020. Please note that the revision deadline will expire at 00.00am on this date. If you do not think you will be able to meet this date please let me know immediately.

Kind regards,

Anita Kristiansen
Editorial Coordinator

on behalf of Dr Kristina Sefc (Associate Editor) and Kevin Padian (Subject Editor)
openscience@royalsociety.org

Associate Editor Comments to Author (Dr Kristina Sefc):

Comments to the Author:

Dear authors,

I am happy to inform you that the three reviewers are very positive about your revision of the manuscript, and would like to ask you to attend to their remaining concerns: Reviewer 1 points out that also juvenile sticklebacks are territorial (not only reproductive males), and reviewer 2 asks you to allow for an alternative explanation of your result, i.e. a difficulty of the fish with the particular learning task. Please include this consideration in the discussion, if deemed appropriate.

Sincerely,

Kristina Sefc

Reviewer comments to Author:

Reviewer: 3

Comments to the Author(s)

I am happy with the revision. The authors have taken into account my comments and those of the other reviewers. There is one minor point that the authors may want to include in the discussion. Reviewer #2, comment 4 and 5, suggested the importance of territorial defense. The authors denied such behaviour in sticklebacks other than reproductive males. This is not correct. Juvenile (males and females) three-spined sticklebacks may defend territories (Bakker, T. C. M. & Feuth-de Bruijn, E. (1988). Juvenile territoriality in stickleback *Gasterosteus aculeatus* L. *Anim. Behav.* 36: 1556-1558.). Juveniles and adult females may also be aggressive towards conspecifics and one can artificially select for higher and lower aggression levels (Bakker, T. C. M. (1986). Aggressiveness in sticklebacks (*Gasterosteus aculeatus* L.): a behaviour-genetic study. *Behaviour* 98: 1-144.).

Reviewer: 2

Comments to the Author(s)

I thank you for your revisions. I'm sorry for late to check your paper because of the influence of such situation around the world. You suggest your results showed that stickleback is not capable of individual recognition because of fission-fusion social system. But males make a territory in breeding season, is there dear enemy phenomenon? Can they modify their brain structure according to the season? And it can't conclude that stickleback is not capable of individual recognition through this experimental procedure. I think these results just showed it is difficult

for them to learn to associate food with another individual or they are not capable of individual recognition during non-breeding season.

Reviewer: 1

Comments to the Author(s)

The authors have clarified my concerns about individuals not learning about the conspecifics due to them already representing a better reward in comparison to the food. They have added some additional information in the discussion about this and have reworded the last couple of sentences in the abstract. I appreciate that the data is already more than 10 years old and more detailed analysis couldn't be conducted.

Overall I do really like the study, the set-up and design is very neat and the results are very clear. I believe the changes in the abstract and the addition in the discussion are appropriate and I would recommend the study for publication.

Author's Response to Decision Letter for (RSOS-191703.R1)

See Appendix B.

Decision letter (RSOS-191703.R2)

Dear Dr Webster,

It is a pleasure to accept your manuscript entitled "No evidence for individual recognition in threespine or ninespine sticklebacks (*Gasterosteus aculeatus* and *Pungitius pungitius*)" in its current form for publication in Royal Society Open Science.

Due to rapid publication and an extremely tight schedule, if comments are not received, your paper may experience a delay in publication. Royal Society Open Science operates under a continuous publication model. Your article will be published straight into the next open issue and this will be the final version of the paper. As such, it can be cited immediately by other researchers. As the issue version of your paper will be the only version to be published I would

advise you to check your proofs thoroughly as changes cannot be made once the paper is published.

on behalf of Dr Kristina Sefc (Associate Editor) and Kevin Padian (Subject Editor)
openscience@royalsociety.org

Appendix A

Dear Editors,

Thank you for giving us the opportunity to revise our manuscript. The referee's comments have been very useful in helping us to produce a clearer paper. We have made most of the changes / clarifications requested by the reviewers, with a few exceptions. These are briefly detailed here and expanded upon below.

(1) Referee 1 provides an alternative explanation for our findings, that the fish displayed a lack of interest in the training procedure- I do not find this plausible for the reasons laid out below and have opted now to pursue it in the discussion. I have performed an analysis for short term anticipatory behaviour similar to the one requested by the referee, but saw no effect. I can include this in the MS if the editor feels it necessary but would prefer not to as it is not something we had made predictions about before running the experiments.

(2) Referee 2 suggests an performing a more precise analysis of the location of the focal fish to each stimulus fish than we currently use (tracking the proportion of time the focal fish is 5 cm from the stimulus fish itself rather than within 5 cm of the enclosure holding the stimulus fish, as we currently do). I agree that this would certainly yield more precise measures but do not think it provides a more accurate measure of recognition, since this is inferred from the time spent with the rewarded stimulus fish relative to the unrewarded one, and the degree of any error associated with the estimates of these times should be equal between the two stimulus fish. In any case, I unfortunately cannot access to the external hard drive containing the trial videos due to the ongoing COVID-19 lockdown.

Full responses to the referees' helpful comments are provided below.

Yours sincerely, Mike Webster, School of Biology, University of St Andrews

Reviewer: 1

Comments to the Author(s)

The study investigates the ability of two species of fish for individual recognition. Replicates of 30 individuals of both species were trained to associate a green light with the provision of a food reward close to either an artificial landmark (blue or yellow cube) or one of two conspecifics in adjacent tank compartments. In the test, both species spend significantly more time near the rewarded landmark when the light was on, however, not next to the rewarded conspecific. The authors conclude that both species can learn the training procedure but are unable to individually recognise specific individuals.

The manuscript is well written, the methods are clear and the analysis appropriate, however I have one concern which needs clarification and possibly additional data:

The presence of conspecifics has been often shown to be an excellent reward in training individual fish and can also outweigh the importance of food rewards (especially when fish are readily fed twice a day 5 bloodworms). I believe it is difficult to tease apart if the fish with the conspecific stimuli didn't learn the association with a specific conspecific due to the lack of individual recognition or if they didn't show 'interest in the training' procedure as they had the constant reward of conspecifics nearby on both sides which might outweigh the food reward presented when the light was on. It is mentioned in the results that both species spend more time near the conspecific stimuli both when the light was on and off due to shoaling and not due to the prospect of the food reward. If fish had learned an association of the light with the food reward I would expect a change in behaviour, even though they haven't learned the correct conspecific stimuli? It would be good to look at specific behaviours when the light is on either during training for the first 60 seconds or over the whole 5 min during the test and compare them with behaviours present in the first 60 seconds training/5min test with the light on in the artificial stimuli treatment. Fish might show anticipatory behaviour of food if they did learn to associate the food with the light, e.g. faster swimming speed, swimming closer to the water surface, closer to the position where food is presented. In the present format there is no strong evidence of a lack of individual recognition but possibly a lack of interest in the training procedure.

A: It is often harder to prove the lack of an effect (impossible?) than its presence. I am not convinced by this alternative explanation (that the fish with the conspecific stimuli didn't learn the association with a specific conspecific... ..as they had the constant reward of conspecifics nearby on both sides which might outweigh the food reward presented when the light was on). Over three weeks the test fish most probably became accustomed to the stimulus fish, whereas the food was only present for short periods of time, and was surely therefore a greater motivator.

These experiments were run more than 10 years ago and I do not have access to the minute by minute time allocation by the test subject to each stimuli fish / landmark. I do however have the first choice data (first stimulus fish / landmark approached). I ran an analysis of first choice using a repeated measures GLM with a binary response variable (rewarded / non-rewarded stimulus visited first)- we saw no evidence that the test subject anticipated the reward by swimming towards the rewarded conspecific or landmark first. I have not included this analysis in the revision, as it was not something we had made predictions about beforehand, but a summary is presented here:

Analysis: First choice

We saw no effect of testing period ($X^2 = 0.07$ df= 112 $P=0.88$), treatment ($X^2 = 0.12$, $P=0.61$) nor any significant interaction between the two ($X^2 = 0.45$, $P=0.24$). We saw no differences between the two species ($X^2 = 0.81$, $P=0.56$), nor any interaction between species and testing period or treatment ($X^2 =$

0.55, $P=0.45$; $X^2= 1.01$, $P=0.20$). Testing period order had no effect upon test subject behaviour ($X^2= 0.05$, $P=0.68$). Nor did the side of the rewarded stimulus ($X^2= 0.51$, $P=0.70$).

We were careful in titling the ms to emphasise 'no evidence' rather than definitively 'no effect' and I have reworked the abstract and discussion to make this clear. It is possible that these fish are capable of individual recognition, but that our experimental design was not sufficient to capture evidence of this- but I can't think of a better way of doing it.

Some minor questions:

Line 187. Why were the fish trained for 21 days? Is there evidence that this is the time they need for associative learning? Have fish been tested in between to see if they learned the association sooner?

A: There is evidence that they can learn these preferences within 1 week (van Bergen, Y., Coolen, I. and Laland, K.N., 2004. Nine-spined sticklebacks exploit the most reliable source when public and private information conflict. *Proceedings of the Royal Society of London. Series B: Biological Sciences*, 271(1542), pp.957-962.), but we had no idea how long individual recognition, if present, might take to acquire. Evidence in other species suggests days to weeks. We reasoned that if recognition was not acquired within a period as long as three weeks then it seemed unlikely that it would occur at all, given how frequently shoals meet and mix in the wild.

Line 187-188. Why was this colour chosen?

A: Previous work in our lab has shown that these fish can learn to associate these colours with a food reward (Pike, T.W. and Laland, K.N., 2010. Conformist learning in nine-spined sticklebacks' foraging decisions. *Biology letters*, 6(4), pp.466-468.). We have added this information to the MS.

In the discussion it was mentioned that sticklebacks have been shown to prefer individuals under better-condition and health as well as larger individuals over smaller ones (Lines 387-392). Is there data available about the condition of the stimuli fish (health, change in size or weight) over the time period of the study? A change in these conditions could maybe have impacted on the ability of the individuals to learn the association as more salient/preferred cues are contradicting it?

A: Unfortunately we do not have the data to test this. I can imagine that if the stimulus fish changed condition at different rates within pairs then this might have affected the test subjects shoaling preferences, but all were fed to satiation each day so this seems unlikely.

Reviewer: 2

Comments to the Author(s)

Sorry for being late, I found this study very interesting, therefore it took long time to read and understand. I have many questions and if I just didn't understand, apologize first.

1. There is no information that which sex was used as subjects and stimulus fishes. In no reproductive individuals, did not they have different interactions between males and females or among same sex?

A: Sticklebacks breed only seasonally and we note in the section 'Collection and housing' that we only used fish that were not in reproductive condition.

2. Fishes were from same location but divided for one month in different aquariums. So, I ask that were the pairs of subjects and stimulus fishes from same aquariums or not, is there any influence on their behavior because of it and how was their interactions before the experiments if they were from same, did they have ranking in the aquarium?

A: A good point- within each tank the test subject and two stimulus fish were each drawn from separate holding tanks and were initially unfamiliar to one another. This information has now been added to the methods.

3. Did subjects have enough motivation to get rewards as soon as possible? If stimulus fishes as themselves have stronger value rather than rewards, I think this experiment need to design again. Could subjects that did not get close to a rewarded stimulus fish at the moment green light was switching take rewards or get extra food after experiments?

A: The food was provided close to the rewarded stimulus fish. These species are facultatively social, and in the wild occur both alone and in groups, so while social contact is almost certainly rewarding I don't expect that it would outweigh a food reward. Also the stimulus fish were present throughout the training and testing period, whereas food was provided only at fixed periods.

4. Results showed subjects were always interested in both stimulus fishes, but there is no description of details in their social interactions. Getting closer includes some meanings and not only for shoaling. I think the behavior of stimulus fishes influence on the behavior of subjects. I ask to show their social interactions during experiments periods, is there any displays, aggressiveness, or modification?

A: When not in breeding condition (which these fish were not) they are not territorial. They can sometimes compete aggressively over food but in this case they were not able to do so directly because they were held in separate compartments. Typically, under these conditions, social attraction and shoaling behaviour are the only interactions seen.

5. Fishes, for example Medaka and Guppy, sometimes have a personal space just like a territory when it is bred in an aquarium alone or with few conspecifics. In this study, there is a possibility that subjects felt high threat-level against stimulus fishes that sometimes changed their location. A previous research in *Neolamprologus pulcher* (an African cichlid) revealed that territorial owners feel high threat against the neighbors that changed their location after dear enemy was established (Sogawa et al. 2016). If so, these results did not mean stickle backs are incapable of individual recognition, these results probably mean they just had stronger motivation to watch out strange neighbors than the rewards.

A: As discussed above, these fish only form territories during the breeding season, and it is only the males that do this. These fish were tested outside of the breeding season and none of the fish displayed male reproductive colouration. I am not aware of any cases documenting these fishes establishing territories for other reasons or of them guarding specific areas for any reason, and I have not seen such behaviour previously. I have no reason to believe that the lack of any preference for associating with the rewarded fish could be explained by territoriality.

6. In this study, the time subjects spent within 5 cm of the barrier was measured. There is no problem in landmark test by this measurement, because landmarks did not change their position. In contrast, stimulus fishes were, probably, always moving. In this case, the results should be measured as the time subjects spent within 5 cm of a stimulus fish. Because if subjects could associate rewards with stimulus fish, the distance of them would become important.

A: This is a good point, and ideally I would track the position of the subject relative to the stimulus fish. The tracking technology sufficiently advanced to this ten years ago when the experiments were run was not available (or at least not affordable) at the time, though in principle it might be possible now. The videos are currently on a hard drive in my office and I cannot obtain them currently because of the COVID-19 lockdown in the UK, so unfortunately I cannot explore this further. In any case I do not think that this could have biased the results because the same measurement imprecision applies both to time spent close to the rewarded and to the unrewarded stimulus fish and there is no reason to expect that the degree of error associated with either measure should differ.

7. I think that even if subjects clear this test, it can't conclude stickleback is capable of individual recognition. There is a possibility that subjects just distinguish rewarded fish from unrewarded fish as a result of class level recognition. To show stickleback is capable of individual recognition, they must identify each rewarded fish. I think this study need to discuss this type of individual recognition called True Individual Recognition.

A: We have concluded here that our study provides no evidence for individual recognition, which is consistent with our findings.

A: I am not sure that I follow the reviewer's argument anyway- pairs stimulus fish were randomly assigned to each test subject and the rewarded and unrewarded stimulus fish were also assigned at random. Rewarded or unrewarded status is not a property or class pertaining directly to the fish themselves (as are things like size, condition, colouration etc), and reward was not paired with any physical characteristic of the stimulus fish in any consistent way. Had the test subjects been shown to be capable of recognising the rewarded fish this could only occur through the association of traits pertaining to that individual with the reward, which could be explained through individual recognition.

8. I also think that it can't conclude that stickleback is incapable of individual recognition or they could not associate rewards with stimulus fishes. I think previous studies do not support former conclusion, latter is easier to explain these results. Because fishes in the wild usually have opportunity to associate food with location or landmark, but rare to associate food with individuals.

A: We have concluded here that our study provides no evidence for individual recognition, and I have changed the abstract to reflect this. I am unaware of the previous studies that the referee is referring to. The Milinski paper that we cite claims this species is capable of individual recognition, but this claim is questionable and other explanations are more plausible (we have tried to state this diplomatically in our discussion). The reviewer is right that our experimental set up differs greatly from what the fish would experience in the wild. This is not a problem though, since our aim was to try and identify a capacity for individual recognition in the first instance. Had we found such a capacity we could have followed it up with further work to establish ecological relevance.

9. There is good discussion about chemical cues of fishes, but no discussion about individual recognition by sight. Some fishes are known that they can identify another by face pattern (e.g. Kohda 2015). Is there any previous study about individual recognition in stickleback by sight?

A: We cite a study that demonstrates class level recognition via visual cues and sticklebacks (Sumpter et al. 2008). We now also cite Siebeck et al. 2010; Kohda et al. 2015; Wang & Takeuchi 2017 as examples of individual recognition by sight in other fish species.

Reviewer: 3

Comments to the Author(s)

This is a well-written, interesting manuscript about the lack of individual recognition in three-spined and nine-spined sticklebacks. It adds significantly to the literature of individual recognition in fishes. I enjoyed reading it and have some suggestions for improvement.

line 179: the green light is introduced here quite suddenly. The reader must be introduced about the reasons for using green light first. Add data on light intensity and light spectrum.

A: The purpose of the green light is introduced in the first sentence of the subsection 'Training' ("We used an experimental design in which test subjects were trained to associate green light with the provision of a food reward close to one of two stimuli."). I have added details on spectrum and intensity as requested.

line 183: this is again unclear. In previous publications of the authors, fish were trained with a distinct green light stimulus but in the present manuscript it is the green illumination of the tank that served as stimulus, which maybe a more difficult learning task as test fish have to discriminate under green light conditions, especially between the two stimulus fish; the colour difference between the landmarks is huge in comparison even under green lighting. The association between award and illumination may be less evident than when the stimulus is a clear separate cue. This should be discussed. Mention how the tank was illuminated when the green light was not on.

A: I have added the following text to the discussion: "It is possible that the landmarks were easier to discriminate than the conspecifics under the green light. This alone though does not seem a compelling explanation for the subjects' lack of any preference for the rewarded conspecific, first because they had ample opportunity to inspect the stimulus fish before the green illumination was presented and second because chemical cues are known to be very important for class-level recognition in these species..."

I have also added information about standard illumination, when the green light was not on.

line 204: netting is not the best way to transport fish and will cause stress even if not noticeable, or did you train fish to the netting?

A: They were not trained to swim into the net, but they had 17 hours between being netted and the next days' testing to recover. All fish fed and behaved normally so I have no reason to believe that there were any long lasting effects of being moved in the net.

Results: please test for side (left, right) bias; often there is a preference for one or the other side that must be controlled for.

A: Side bias was included in a new GLM- there was no evidence of a side bias. This model is reported in the revised main text.

Please also test whether fish preferred one colour over the other.

A: Colour bias was also investigated- there was no evidence of any colour bias among the fish in the landmark treatment. This is also reported in the revised main text.

Label the y-axis in Fig. 2.

A: Corrected

line 291: "green light stimulus" is confusing as it was actually a green illumination

A: Corrected

line 311: you did not measure in the ultraviolet waverange that sticklebacks can perceive well. This should be discussed; there are fish that use UV signals for (species) recognition (Siebeck et al. 2010. Curr Biol 20:407-410). A paper on individual recognition based on visual cues in medaka was not cited (Wang & Takeuchi 2017. eLife 6:e24728).

A: I have now cited these papers. I have opted not to include discussion of UV signals, as signalling is a little off topic for our discussion.

line 347: write "split"

A: Corrected

line 474, 476, 479: write "T.C.M." instead of "T.C."

A: Corrected

References: write species names in italic

A: Corrected

Appendix B

Dear Dr Sefc,

Thank you for giving us the opportunity to revise our paper, and thank you to the referees for their helpful comments. We are pleased that reviewers are satisfied with the changes that we made. Reviewer 1 requested no further changes, while Reviewer 2 asked about (1) the *Dear Enemy* effect in these species, and (2) alternative interpretations of our findings. As we have outlined in our response below, there are (1) no studies that we are aware of that demonstrate the *Dear Enemy* effect in sticklebacks. I have argued below that (2) our interpretation of our findings is sufficiently cautious and that the reviewer's suggestions are already considered in the first paragraph of our discussion. Finally, Reviewer 3 points out that contrary to our claims in our last response to reviewers document, there is in fact evidence of territoriality in non-breeding sticklebacks. We were unaware of this work and thank the reviewer for pointing it out. As we've argued below though, it is not closely relevant to the interpretation of our findings and so we have not discussed this work in our manuscript. Given these responses, we have made no further changes to the text of the manuscript.

Yours Sincerely,

Mike Webster

Reviewer: 1

The authors have clarified my concerns about individuals not learning about the conspecifics due to them already representing a better reward in comparison to the food. They have added some additional information in the discussion about this and have reworded the last couple of sentences in the abstract. I appreciate that the data is already more than 10 years old and more detailed analysis couldn't be conducted.

Overall I do really like the study, the set-up and design is very neat and the results are very clear. I believe the changes in the abstract and the addition in the discussion are appropriate and I would recommend the study for publication.

Thank you.

Reviewer: 2

I thank you for your revisions. I'm sorry for late to check your paper because of the influence of such situation around the world. You suggest your results showed that stickleback is not capable of individual recognition because of fission-fusion social system. But males make a territory in breeding season, is there dear enemy phenomenon? Can they modify their brain structure according to the season? And it can't conclude that stickleback is not capable of individual recognition through this experimental procedure. I think these results just showed it is difficult for them to learn to associate food with another individual or they are not capable of individual recognition during non-breeding season.

Thank you for your comments. To address the first point, there is evidence that territorial males habituate to the presence of a neighbouring territory holder, but I am aware of no study that has demonstrated that they learn the identity of their neighbouring rivals. Such a study might allow a focal male to habituate to rival A and then switch A for rival B- if the focal male's aggressive response is 'reset' (and appropriate controls are carried out) then we might conclude that the dear enemy

effect is occurring, which implies recognition. I'm not aware of any study that has specifically included these conditions, however.

Second, we don't state that sticklebacks are not capable of individual recognition, we are careful to only claim no evidence for this, which I think is reasonable. It is difficult, perhaps impossible, to show that a behaviour doesn't exist, and one day a study might provide compelling evidence for individual recognition in this species. I am satisfied that we are sufficiently cautious not to overstate our findings in the paper (e.g. first paragraph of the discussion).

Reviewer: 3

I am happy with the revision. The authors have taken into account my comments and those of the other reviewers. There is one minor point that the authors may want to include in the discussion. Reviewer #2, comment 4 and 5, suggested the importance of territorial defense. The authors denied such behaviour in sticklebacks other than reproductive males. This is not correct. Juvenile (males and females) three-spined sticklebacks may defend territories (Bakker, T. C. M. & Feuth-de Bruijn, E. (1988). Juvenile territoriality in stickleback *Gasterosteus aculeatus* L. *Anim. Behav.* 36: 1556-1558.). Juveniles and adult females may also be aggressive towards conspecifics and one can artificially select for higher and lower aggression levels (Bakker, T. C. M. (1986). Aggressiveness in sticklebacks (*Gasterosteus aculeatus* L.): a behaviour-genetic study. *Behaviour* 98: 1-144.).

Thank you, I was unaware of these findings (which I ought to have known, since I have of course read Bakker 1986). As we saw no evidence of aggression in our study we think that it is unlikely to explain our findings here. As our paper is already quite long and these papers not directly related to our work I have not discussed them.